# Human Milk Feeding Is Associated with Decreased Incidence of Moderate-Severe Bronchopulmonary Dysplasia in Extremely Preterm Infants

**DOI:** 10.3390/children10071267

**Published:** 2023-07-23

**Authors:** Sergio Verd, Roser Porta, Gemma Ginovart, Alejandro Avila-Alvarez, Fermín García-Muñoz Rodrigo, Montserrat Izquierdo Renau, Paula Sol Ventura

**Affiliations:** 1Department of Pediatric Care Primary Unit, La Vileta Surgery, Health Sciences Research Institute (IUNICS), Balearic University, 07122 Palma, Spain; drsverd@gmail.com; 2Department of Pediatrics, Division of Neonatology, Hospital Universitari Germans Trias i Pujol, Badalona, Universitat Autònoma de Barcelona, 08193 Bellaterra, Spain; gginovartgaliana@gmail.com (G.G.); paulasolventura@hotmail.com (P.S.V.); 3Division of Neonatology, Complexo Hospitalario Universitario de A Coruña (CHUAC), SERGAS, 15006 A Coruña, Spain; alejandro.avila.neonatologia@gmail.com; 4Division of Neonatology, Complejo Hospitalario Universitario Insular Materno-Infantil, 35016 Las Palmas, Spain; fgarciamu@gmail.com; 5Division of Neonatology, Hospital Sant Joan de Déu, 07007 Barcelona, Spain; montserrat.izquierdo@sjd.es

**Keywords:** bronchopulmonary dysplasia, human milk feeding, breastfeeding, extremely preterm

## Abstract

Background: An increased rate of bronchopulmonary dysplasia (BPD) is reported in extremely preterm infants. A potential role of human milk feeding in protecting against this condition has been suggested. Methods: A retrospective descriptive study was conducted based on data about morbidity in the population of infants born between 22+0 and 26+6 weeks of gestation, included in the Spanish network SEN1500 during the period 2004–2019 and discharged alive. The primary outcome was moderate-severe BPD. Associated conditions were studied, including human milk feeding at discharge. The temporal trends of BPD and human milk feeding rates at discharge were also studied. Results: In the study population of 4341 infants, the rate of moderate-severe BPD was 43.7% and it increased to >50% in the last three years. The factors significantly associated with a higher risk of moderate-severe BPD were birth weight, male sex, high-frequency oscillatory ventilation, duration of invasive mechanical ventilation, inhaled nitric oxide, patent ductus arteriosus, and late-onset sepsis. Exclusive human milk feeding and any amount of human milk at discharge were associated with a lower incidence of moderate-severe BPD (OR 0.752, 95% CI 0.629–0.901 and OR 0.714, 95% CI 0.602–0.847, respectively). During the study period, the proportion of infants with moderate-severe BPD fed any amount of human milk at discharge increased more than twofold. And the proportion of infants with moderate-severe BPD who were exclusively fed human milk at discharge increased at the same rate. Conclusions: Our work shows an inverse relationship between human milk feeding at discharge from the neonatal unit and the occurrence of BPD.

## 1. Introduction

Bronchopulmonary dysplasia (BPD) is one of the most severe consequences of extremely preterm birth. Advances in many non-nutrition-related aspects of care, particularly avoiding invasive ventilation [1] and fluid overload [2], using caffeine [3], and late systemic corticosteroids [4] in those infants unable to go off of mechanical ventilation are evidence-based strategies to lower the potential for BPD development. However, contrary to documented decrease in most neonatal morbidity, the incidence of BPD remains unchanged [5] or is increasing [6,7,8].

On the other hand, it has been reported that for every ten percent increase in human milk (HM) feeding in preterm neonates, the risk of BPD is lowered by 9.5% [9] and that a daily intake of 50 mL/kg/day of HM during the first month of life can minimize the incidence of BPD [10]. Aiming to summarize the results of previous research on this topic, a meta-analysis found that feeding exclusively mothers’ milk might reduce the incidence of BPD [11]. Another meta-analysis found a significantly lower incidence of BPD in donor milk (DM)-fed preterm infants compared to their formula-fed counterparts [12]. BPD pathophysiology is still not completely understood, but there is an agreement to define it as a multifactorial disease in which a combination of oxidative stress and inflammation harms the premature infant’s growing lungs that are particularly vulnerable to injury [13]. Among all antioxidant therapies and strategies, HM is one of the most straightforward and accessible. HM is rich in antioxidants, which play a major role in maintaining the equilibrium between oxidative stress and antioxidant systems [14,15].

Therefore, promoting mother’s own milk (MOM) is one of the priorities of modern neonatal care. An estimated 75% of high-risk Spanish neonates have access to DM [16]. The option of giving DM when MOM is unavailable or not sufficient to cover infant needs is almost always present in Spanish neonatal intensive care units (NICUs).

In light of the above, we considered it very important to focus on the contribution of HM feeding to the lower risk of BPD, the paucity of studies on this strategy in preterm infants at the limit of viability, and the fact that in many countries, there is a large availability of DM in case there is not enough MOM.

In this context, we undertook an observational cohort study to further explore the relationship between HM feeding and BPD rates in extremely preterm infants in recent years in Spain.

## 2. Materials and Methods

### 2.1. Study Design

A retrospective descriptive, multi-hospital cohort study was designed based on data obtained prospectively from the Spanish neonatal network SEN1500, a database created in 2002.

### 2.2. Population

SEN1500 collects the perinatal data on the morbidity and mortality of premature infants born with a birthweight less than 1500 g, either born at the 64 collaborating centers or admitted to them in the first 28 days of life [17]. SEN1500 includes premature infants born alive from 22+0 weeks. The population represents approximately two-thirds of extremely preterm infants born in Spain. The list of researchers and hospitals from the SEN1500 Neonatal Network is detailed in a Appendix A.

From our database, 8739 extremely preterm infants from 22+0 to 26+6 weeks born between 2004 and 2019 were initially recruited in this study. Those who died before discharge (47.5%), those with incomplete data in the primary interest variable (2.4%), or with severe congenital anomalies (2.8%) were excluded. According to this selection, 4341 infants were finally included for analysis (Figure 1).

Legend of Figure 1: Flow chart from the starting population consisted of all preterm infants born below 27 weeks between 2004 and 2019 registered in SEN1500. The study sample is obtained after excluding the deceased, those with major congenital anomalies, and those with incomplete or missing data related in the primary interest variable.

### 2.3. Variables

We aimed to explore the clinical factors related to the development of moderate-severe BPD (variable of interest), focusing mainly on the presence or absence of HM feeding (exclusive or any amount of HM feeding) at discharge. Moderate-severe BPD was defined as the need for supplementary oxygen or positive pressure at 36 weeks of postmenstrual age (PMA) [18]. The classical criteria for BPD consider it as moderate if the infant is under supplementary oxygen with a FiO2 less than 0.3 and severe if the FiO2 is more than 0.3, or any positive pressure at 36 weeks of PMA. Positive pressure included invasive mechanical ventilation (iMV), nasal continuous airway pressure (nCPAP), nasal intermittent positive pressure (nIPPV) or high-flow oxygen cannulae (HFOC). Demographic and clinical factors were assessed for confounding, testing for any association between an independent variable and BPD (dependent variable) occurrence. Independent variables were selected from among those considered as of possible importance for the short-term outcome. The maternal and delivery characteristics were gestational age (GA) estimated in weeks based on the last menstrual date and ultrasound calculation, multiple pregnancies, in vitro fertilization (IVF), place of birth (inborn or outborn), antenatal steroids (complete or partial course), cesarean section, maternal chorioamnionitis (defined by clinical criteria, positive amniotic fluid culture [19], and placental culture or histological exam when available), and maternal hypertensive disorder. The infant demographic or clinical characteristics are listed in Table 1: sex, Apgar score at 1′ and 5′, birthweight, clinical index risk for babies (CRIB) score [20], use and duration (hours) of invasive mechanical ventilation, high frequency oscillatory ventilation (HFOV), non-invasive mechanical ventilation with nasal continuous positive pressure (nCPAP) or nasal intermittent positive pressure (nIPPV), inhaled nitric oxide (iNO), surfactant administration, prophylactic indomethacin, postnatal systemic steroids, early-onset sepsis, late-onset sepsis (LOS) defined as suggestive clinical symptoms along with positive blood culture after 72 h of life, need for inotropic therapy, necrotizing enterocolitis (NEC) Bell’s stage II or more [21] and NEC requiring surgery, patent ductus arteriosus (PDA) confirmed by echocardiography and managed with medical or surgical treatment, major cerebral lesion defined as intraventricular hemorrhage (IVH) grade 3 or more according to the Papile criteria [22] and/or periventricular leukomalacia (PVL) defined as the presence of changes in periventricular white matter detected by ultrasound 3 weeks after birth, retinopathy of prematurity (ROP) grade 3 or more according to the ROP International Classification [23], or ROP surgically treated, and length of admission in the NICU.

*Feeding issues.* Each neonatologist recommended a feeding type based on the infant’s maturity, HM availability, and nutritional needs. The preterm formulae used in the neonatal units during this period were enriched in calcium, phosphorus, protein, and calories.

### 2.4. Ethics Requirements

The Ethics and Research Committee of the Hospital Universitari Germans Trias i Pujol approved the study on date 11 March 2021 (PI-21-051). The SEN1500 data registry is anonymous. Ethics and research committees of the collaborating centers have adopted data protection rules and regulations.

### 2.5. Statistical Analysis

The data were expressed as means (SD) for continuous variables or as numbers/percentages for categorical variables. The normality of distribution and equality of variances were evaluated through the Kolmogorov–Smirnov and Levene’s test, respectively. The Student’s T test or the Mann–Whitney test were used to compare independent variables according to their size and characteristics between the two groups. The ANOVA or Kruskal–Wallis non-parametric tests were used to compare differences among more than two groups. A Pearson chi-square test was used to compare categorical variables between groups. The odds ratio and 95% confidence intervals were calculated. To examine more complex relationships between variables, multivariate analysis was performed using logistic regression to model the probability of dichotomous events (moderate-severe BPD, yes/no). Regression explicative models were built based on previous theoretical evidence and explanatory variables correlated with moderate-severe BPD detected in the univariate analysis. The first model examined the associations between explanatory variables and moderate-severe BPD, not including HM feeding at discharge. The second examined the associations among explanatory variables, including exclusive HM feeding at discharge and moderate-severe BPD. The third includes “mixed” (any amount of HM feeding at discharge) instead of exclusive HM feeding. The statistical significance level in the multivariate analysis was *p* < 0.05 for a two-tailed test. Nagelkerke’s R-squared values were used to describe the variability explained by each model. The Statistical Package for Social Science SPSS version 23.0 was used to perform all the statistical analyses, establishing the significance at *p* < 0.05.

## 3. Results

The results align with the main objective of this study, which was to explore the relationship between the type of feeding at discharge from the neonatal unit and the occurrence of BPD in preterm infants at the limit of viability.

### 3.1. Selection of Study Cohort

Data from 8739 infants born at a GA of 22+0 to 26+6 weeks was registered in the database during the study period. Figure 1 shows the flow chart of the study population. After excluding 246 infants with severe congenital anomalies, 209 infants with incomplete or missing data in the primary interest variable, and 3943 infants who died before discharge, those 4341 infants who survived to discharge were included for analysis.

The rate of moderate-severe BPD in the entire population was 43.7%. A significant increasing trend was observed during the study period (Figure 2).

Legend of Figure 2: Historical analysis of our population. Occurrence from 2004 to 2019 of moderate-severe BPD among preterm infants under 27 gestational weeks from the SEN1500 database that survive until discharge from the NICU.

### 3.2. Infants’ *Characteristics*

Table 1 shows the characteristics of the total population and the differences between those with moderate-severe BPD (n: 1897, 43.7%) and those with no moderate-severe BPD (n: 2444, 56.3%). On average, the gestational age of neonates in our sample was 25.7 (±0.84) weeks of gestation, and their birthweight was 812.5 g (±151.0). In total, 47.9% of them were girls. The rate of multiple births was 27.4% and 4.5% were outborn.


*The bivariate analysis identifies risk factors associated with moderate-severe BPD.*


The results of the univariate analysis showed an inverse association of GA, birth weight, and Apgar score at 1′ and 5′ with moderate-severe BPD. On the other hand, the bivariate analysis showed that the factors associated with moderate-severe BPD were male sex, CRIB score, iMV, HFOV, hours of iMV, iNO, surfactant administration, prophylactic indomethacin, postnatal systemic steroids, LOS, inotropic therapy, NEC and surgical NEC, PDA, major cerebral lesion, and ROP. In addition, the percentage of infants receiving exclusive HM feeding after discharge was significantly higher in those free of moderate-severe BPD (39.1% versus 30.3%, *p* < 0.001; OR 0.675, 95% CI 0.593–0.768). Also, 63.6% of infants free of moderate-severe BPD received any amount of HM feeding after discharge versus 58.5% of infants with moderate-severe BPD (*p* < 0.001; OR 0.607, 95% CI 0.537–0.698). A multivariate analysis was subsequently performed (Table 2). In the regression models, variables associated with moderate-severe BPD detected in the previous univariate analysis and considered technically crucial in the development of BPD, according to scientific evidence, were included. In the second and third models, adding HM feeding increased the explained variance from 0.32 to 0.33 (Nagelkerke’s R-squared). Exclusive HM feeding after discharge was associated with a lower incidence of moderate-severe BPD (OR 0.758, 95% CI 0.641–0.896), as any amount of HM feeding after discharge did (OR 0.770, 95% CI 0.656–0.904).

Finally, in infants with moderate-severe BPD, exclusive HM feeding after discharge increased from 14.4% in 2004 to 36.4% in 2019 (*p* < 0.001), as shown in Figure 3, and any amount of HM feeding after discharge increased from 26.7% in 2004 to 60.8% in 2019 (*p* < 0.001).

Legend of Figure 3: Proportion of extremely preterm infants with moderate-severe BPD receiving HM feeding at hospital discharge from 2004 to 2019.

## 4. Discussion

In this multi-center national cohort study on the morbidity of extremely preterm infants, we found that those discharged while fed HM were at a lower risk of moderate-severe BPD than those receiving no HM after discharge.

### 4.1. The Changing Epidemiology of BPD over the Study Period

Also, a significant increase in BPD rates and the percentage of HM feeding at discharge over the period was observed. This improvement in our HM provision might be related to implementing developmental and family-centered care in Spanish units over the study period [24]. In recent years of the study, less than half of the infants in our population were free from moderate-severe BPD. Most studies on temporal trends in respiratory outcomes of extremely preterm infants have reported similar results [5,6,7,8]. Despite introducing less iMV strategies, the rates of BPD have kept unchanged or even worsened in the more immature infants. Several reasons could explain this finding. First is the changes in the definition and management of BPD and its non-homogenous condition. The classical definition of moderate-severe BPD was based on the need for supplementary oxygen therapy or any positive pressure at 36 weeks of PMA. More recently, according to the mode of respiratory support administered at 36 weeks of PMA, regardless of supplemental oxygen use, an updated definition of BPD has been proposed [25]. The definition of moderate-severe BPD (or grade 2 or 3) according to the new criteria is the need for nCPAP or HFOC ≥ 2 L/min (moderate) or iMV (severe). The SEN1500 database started introducing data on any positive pressure ventilation at 36 weeks of PMA in 2010 when nCPAP and HFOC became generalized in Spanish NICUs. Second, an increase in survival of infants born at extreme GA has been suggested to explain the increase in morbidity in survivors [6]. Survival of infants born at 23+0 to 26+6 weeks has increased over the study period in Spain at all GA, from less than 50% to more than 60% [26]. A trend to a more active perinatal approach in those born at 23+0–23+6 weeks has been observed in the last decade, but morbidity has remained unchanged [27]. The more active the approach to resuscitation and treatment of extreme preterm infants, the higher the survival rates achieved. Consequently, the rates of moderate-severe BPD in this group of preterm infants rise. This is a possible explanation for our sample’s relatively low rate (43.7%) of moderate-severe BPD. Another possible explanation for the low rate of moderate-severe BPD in our population is the progressive increase in the rates of HM feeding in preterm infants in our country. Moreover, as there is usually no single cause for changes in neonatal morbidity, it is most likely that these two changes have contributed simultaneously to the current rates of moderate-severe BPD. In a comparative study including the EXPRESS (Sweden), the EPICURE-2 (UK), and the EPIPAGE-2 (France) cohort [28], the first two studies reported BPD rates of 68.6% and 77%, respectively, and the last one, with fewer survivors under 25 weeks, a rate of 39.2%. The factors associated with a greater risk of BPD encountered in this study have also been described previously, mainly the severity of the respiratory condition during the first days, the significant PDA, and the LOS [29].

In recent years, there has been a reinforced interest in the two main issues to which this study has been devoted. On one hand, since data are lacking regarding the prognosis of infants born shortly before and after 24 weeks, we must admit that most treatments for these neonates are not yet supported by scientific evidence [30,31]. Moreover, on the other hand, ongoing research on the pathogenesis of BPD analyzes the potential value of HM to prevent and control this neonatal complication [11,12].

Antioxidant activity may be the main mechanism through which HM prevents BPD. Different agents induce BPD, the most common of which can be oxidative stress [32], Among a small number of preventive targets for BPD, the control of oxidative overload stands out as one of the most promising. When the lungs of extremely premature infants are still developing, hyperoxia increases alveolar epithelial cell death and pulmonary vascular remodeling [33]. Even though oxidative imbalance plays a role in the development of BPD, research on antioxidant molecules in preterm infants is limited. As a result, determining how to administer exogenous antioxidant therapy is a delicate matter [34]. Feeding HM to preterm infants is not challenging. It is aligned with the oxidant/antioxidant balance that starts before birth when the fetus’s antioxidant system is upregulated and immediately thereafter when antioxidants are transferred from the mother’s milk to the newborn infant. HM is the safest food for these patients. HM delivers several active compounds and stem cells that are thought to minimize oxidative imbalance in the BPD process [34,35].

### 4.2. Other Likely Mechanisms Underlying the Effects of HM Feeding on BPD

Neonatal nutrition plays a significant influence in modifying lung alveolar growth as well as alveolar surface area. Because most alveolar growth occurs shortly after birth, optimizing nutrition during the postnatal period is critical [36,37]. However, achieving the best balance at such times and for such patients is complicated by the need for a diet based on HM that must be particularly rich in calories.

Immunoglobulins, immunological cells, anti-inflammatory mediators, and the microbiota are all present in HM. In addition, a longer duration of HM has been linked to larger thymic size and higher counts of T lymphocytes beyond infancy [38]. However, the mechanisms to mitigate BPD among HM-fed neonates are elusive.

### 4.3. Lower Risk of BPD among HM-Fed Preterm Infants

Three systematic reviews have analyzed the risk of BPD after HM intake (MOM or DM, both pasteurized or unpasteurized HM) in very low birth weight (VLBW) infants [11,12,39]. These studies have compared exclusive HM intake to exclusive formula intake, any HM intake to exclusive formula intake, and the dose–response effect with HM intake. The reviews quoted above have reported different types of improved respiratory outcomes among HM-fed preterm infants: DM supplementation had a protective effect on BPD and on iMV requirements [12], an exclusive HM diet dramatically reduced the risk of BPD [11], feeding infants with raw HM instead of pasteurized HM protected VLBW infants from BPD [11], and finally, it is confirmed that exclusive, mainly, or any HM feeding shows a protective effect against BPD development [39].

Since these systematic reviews, studies on the effects of HM feeding on the occurrence or severity of BPD have continued to be published. A 2020 retrospective study on 1363 VLBW infants found that those receiving 50 mL/kg/day of HM during the first four weeks of life had lower odds of BPD, as well as of moderate and severe BPD; in contrast, the authors found no effect of 1–24 mL/kg/day or 25–49 mL/kg/day of HM on BPD nor on the degree of severity of the disease [10]. A 2022 retrospective analysis has reported a new finding: the authors focused on the first time that premature infants received HM and showed that the first HM feeding within 72 h of birth is linked to lower prevalence of moderate-to-severe BPD in VLBW infants and that occurrence of moderate-severe BPD was not linked to HM aggregates during the hospital stay [40]. Additionally, a research from 2022 on 4470 very/extremely low-birth-weight infants showed a 31% reduction in BPD incidence among exclusively HM-fed preterm neonates [41]. Finally, according to a 2023 multivariate analysis, the risk of BPD is 66% lower among VLBW infants fed fresh HM +/− preterm formula than among their counterparts fed pasteurized HM +/− preterm formula [42]. Hence, recent research describes nonlinear dynamics and thresholds that can provoke large pathological responses. This kind of models often govern the biological responses.

Since 2011, more than half of the infants below 26 weeks of gestation with BPD in our study population received some HM feeding at NICU discharge. This is notable when compared with previous reports on percentages of preterm infants that continue to receive HM after NICU discharge: 12% of infants below 28 weeks of gestation [43], or 28–50% of VLBW infants in US studies, or 38–66% of VLBW infants in worldwide studies [44,45]; or 13–49% of all preterm infants during the hospital stay in developed countries [46].

While previous studies have reported rates of HM feeding after NICU discharge, limited research specifically provided details of the successive stages of HM discontinuation during the hospital stay of preterm infants. Further, we are unaware of such studies on extremely preterm infants. A recent report on preterm infants (23–36 weeks of gestation) describes that 5.3% of neonates did not receive any HM, 48.7% of them received some HM but stopped prior to NICU discharge, and 46% of them continued some HM after NICU discharge [47]. This raises the question regarding the proportion of HM and formula fed to preterm infants in our population who were exclusively or mixed HM-fed after hospital discharge. We still need to collect detailed feeding information to answer this question. However, given that we know that most preterm infants are fed HM at some point during their stay in the NICU but that about half of them, later in their admission, become exclusively formula-fed, it is not entirely wrong to infer that the three groups of preterm infants in our sample have taken increasing proportions of HM depending on whether they never took HM or stopped taking it during admission, or whether they continued to take some HM or exclusively HM after discharge.

Therefore, our results align with previous research that has shown not only how partial feeding with DM or fresh or frozen MOM milk can minimize the occurrence of BPD but also points to a non-linear and inverse dose–response relationship between proportions or amounts of HM provided and cases of BPD. Our study contributes to increasing the knowledge regarding the potential benefit of DM or MOM in reducing BPD, specifically in the group of more immature infants. The protective role of any amount of breastmilk for infants with BPD continues after discharge. It has been associated with fewer emergency department visits, systemic steroid courses, fewer cough or chest congestion episodes, and a trend toward fewer hospitalizations [48]. Hence, if BPD could be prevented to some degree in the neonatal period, the respiratory health of former extremely preterm infants would be enhanced for years after the neonatal period.

#### 4.3.1. Limitations

This study has some potential limitations. It is an explicit limitation that this descriptive study was not designed to establish cause-and-effect relationships. There is no intervention as this is a retrospective observational study; therefore, the relationship between HM feeding and BPD must be regarded as an association. In addition, data about the duration and amount of HM received from birth to discharge are unavailable. The high drop-out rate of extremely preterm infants from cohorts could affect this study’s results.

As a second limitation, changes in medical practices concerning BPD have been introduced during the study period, primarily related to the use of non-invasive nasal positive pressure and HFOC. The variables nCPAP, nIPPV, and HFOC at 36 weeks of PMA were included in the database in 2010. Some infants discharged before 2010 under HFOC without supplemental oxygen (FiO2 0.21) at 36 weeks PMA could not have been included in the moderate-severe BPD group. Nevertheless, most infants in this condition received a FiO2 greater than 0.21, so they would probably be categorized as moderate-severe BPD, and the use of HFOC in Spanish NICUs was extremely uncommon before 2010.

#### 4.3.2. Strengths

The strengths of the present study are the large number of participating extreme preterm infants, the fact that the study was obtained from a national registry, and that the feeding data have been collected by neonatologists in each NICU through the medical records of each infant after discharge to reduce recall bias.

## 5. Conclusions

The results of the present study support the link between any amount of HM feeding and a lower risk of BPD in the most vulnerable population, the extremely preterm infants.

The quality bundles and strategies implemented in NICUs to prevent and improve outcomes of infants with BPD should always include the provision of HM and the promotion of long-term breastfeeding.

## Figures and Tables

**Figure 1 children-10-01267-f001:**
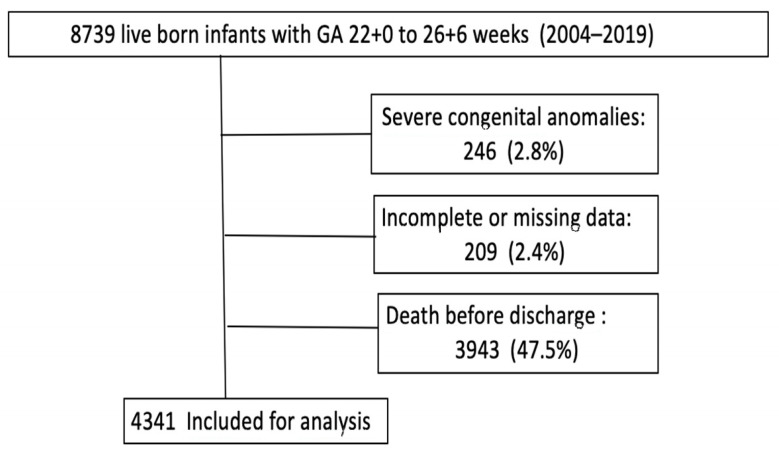
Flow chart of the study population.

**Figure 2 children-10-01267-f002:**
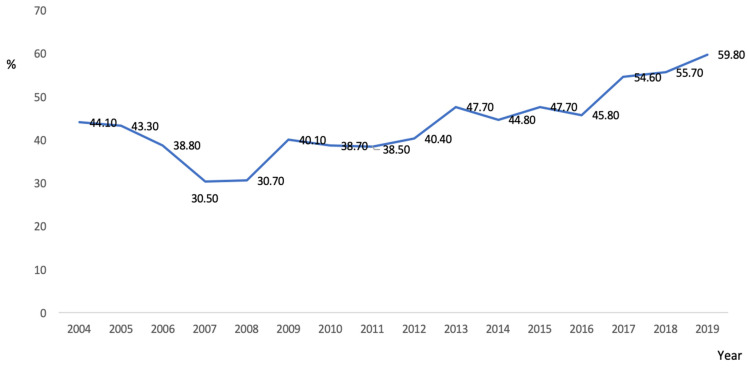
Moderate-severe BPD rates (%) over the study period. *p* < 0.001.

**Figure 3 children-10-01267-f003:**
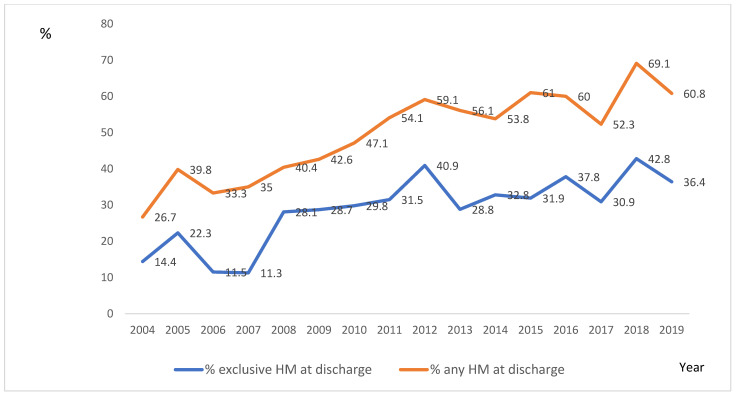
HM feeding at discharge over the study period in infants with moderate-severe BPD.

**Table 1 children-10-01267-t001:** Characteristics of the total population and comparison between infants with moderate–severe BPD and infants with mild or no moderate–severe BPD.

	Total Population n = 4341	Moderate–Severe BPD n = 1897/4341 (43.7%)	No Moderate–Severe BPD n = 2444/4341 (56.3%)	*p*	OR (95% CI)
Gestational age (weeks)	25.7 (0.84)	25.5 (0.900)	25.8 (0.85)	<0.001	
Birth weight (g)	812.58 (151.004)	773.69 (145.975)	842.77 (148.007)	<0.001	
Male sex	2262/4341 (52.1)	1072/1897 (56.5)	1190 /2444 (48.7))	<0.001	1.369 (1.214–1.545)
Multiple birth	1188/4341 (27.4)	548/1897 (28.9)	640/2444 (26.2)	0.048	1.145 (1.001–1.309)
IVF	793/4046 (19.6)	357/1783 (20.0)	436/2263 (19.3)	0.548	1.049 (0.897–1.226)
Antenatal steroids	3889/4341 (89.6)	1694/1897 (89.3)	2195 /2444 (89.8)	0.583	0.947 (0.778–1.151)
Complete antenatal steroids	2846/4341 (65.6)	1242/1897 (65.5)	1604/2444 (65.6)	0.877	
Outborn	195/4341 (4.5)	95/1897 (5)	100/2444 (4.1)	0.148	1.232 (0.967–1.647)
Caesarean section	2343/4341 (54)	1043/1897 (55)	1300 /2444 (53.2)	0.241	1.075 (0.953–1.212)
Chorioamnionitis	1215/3078 (39.5)	562/1397 (40.3)	653/1681 (38.8)	0.437	1.032 (0.954–1.117)
Maternal hypertensive disorder	279/3085 (9)	136/1394 (9.8)	143/1691 (8.5)	0.21	1.170 (0.915–1.497)
1 min Apgar score	5 (4–7) ^a^	5 (4–7) ^b^	6 (4–7) ^c^	<0.001	
5 min Apgar score	8 (7–9) ^d^	8 (6–8) ^e^	8 (7–9) ^f^	<0.001	
CRIB score	4 (2–7) ^g^	5 (4–8) ^h^	4 (2–6) ^i^	<0.001	
iMV	3746/4341 (86.3)	1753/1897 (92.4)	1993/2444 (81.5)	<0.001	2.755 (2.259–3.359)
HFOV	1209/4321 (28)	856/1888 (45.3)	353/2433 (14.5)	<0.001	4.887 (4.229–5.6489
Duration of iMV (hours)	440.73 (554.78)	699.02 (677.1)	242.42 (318.42)	<0.001	
nCPAP	3863/4333 (89.2)	1670/1893 (88.2)	2194/2440 (89.9)	0.074	0.840 (0.693–1.017)
nIPPV	2002/3473 (57.6))	922/1521 (60.6)	1080/1953 (55.3)	0.002	1.244 (1.086–1.426)
iNO	317/4136 (7.7%)	253/1803 (14)	64/2333 (2.7)	<0.001	5.787 (4.366–7.670)
Surfactant therapy	3440/4333 (79.4)	1624/1896 (85.7)	1816/2437 (74.5)	<0.001	2.042 (1.744–2.390)
Prophylactic indomethacin	286/4335 (6.6)	74/1983 (3.9)	212/2442 (8.7)	<0.001	0.428 (0.326–0.562)
Postnatal systemic steroids	1177/4324 (27.2)	851/1892 (45.0)	326/2432 (13.4)	<0.001	5.281 (4.556–6.122)
Early onset sepsis	295/4326 (6.8)	121/1887 (6.4)	174/2439 (7.1)	0.35	0.892 (0.701–1.134)
Late onset sepsis	2773/4339 (63)	1405/1896 (74.1)	1328/2443 (54.4)	<0.001	2.403 (2.110–2.736)
Inotropic therapy	1808/4139 (43.7)	1016/1808 (56.3)	792/2334(33.9)	<0.001	2.507 (2.209–2.845)
NEC	513/4341 (11.8)	277/1897 (14.6)	236/2444 (9.7)	<0.001	1.600 (1.329–1.925)
NEC surgery	379/4335 (8.7)	209/1895 (11)	170/2440 (7)	<0.001	1.655 (1.339–2.046)
PDA medical/surgical treatment	2576/4268 (60.35)	1336/1869 (71.5)	1240/2399 (51.7)	<0.001	2.343 (2.060–2.664)
Major cerebral lesion	872/4341 (20.1)	483/1897 (25.5)	389/2444 (15.9)	<0.001	1.805 (1.554–2.096)
ROP surgery	674/4341 (15.5)	405/1897 (21.3)	269/2444 (11)	<0.001	2.195 (1.856–2.595)
ROP grade 3 or more	685/4244 (16.1)	428/1856 (23.1)	257/2388 (10.8)	<0.001	2.485 (2.100–2.942)
Duration of admission in NICU (days)	99 (30) ^j^	114 (32) ^k^	87 (22) ^l^	<0.001	
Exclusive HM feeding at discharge	1493/4222 (35.2)	559/1851 (30.2)	934/2391 (39.1)	<0.001	0.675 (0.593–0.768)
Any amount of HM feeding at discharge	2474/4242 (58.3)	953/1851 (58.3)	1521/2391 (63.6)	<0.001	0.607 (0.537–0.698)

Qualitative variables expressed by n (%) and *p* value was calculated via chi-square test. Quantitative variables expressed by mean (SD) and *p* value was calculated using *t*-test. Ordinal variables (scores) expressed by median (range) and *p* value was calculated using Mann–Whitney test. col %, column percent. a. Data obtained in 4286/4341; b. data obtained in 1873/1897; c. data obtained in 2413/2444; d. data obtained in 4214/4341; e. data obtained in 1837/1897; f. data obtained in 2377/2444; g. data obtained in 3766/4341; h. data obtained in 1637/1897; i. data obtained in 2129/2444; j. data obtained in 4314/4341; k. data obtained in 1884/1897; l. data obtained in 2430/2444. Abbreviations. IVF: in vitro fertilization, CRIB: clinical index risk for babies; HFOV: high-frequency oscillatory ventilation, nCPAP: nasal continuous positive pressure, nIPPV: nasal intermittent positive pressure, INO: inhaled nitric oxide, NEC: necrotizing enterocolitis, PDA: patent ductus arteriosus, ROP: retinopathy of prematurity, NICU: neonatal intensive care unit, HM: human milk.

**Table 2 children-10-01267-t002:** Multivariate analysis of risk factors for moderate –severe BPD.

	Model 1		Model 2		Model 3	
	‘HM Feeding at Discharge’ Not Included		Including ‘HM Feeding at Discharge’		Including ‘Any Amount of HM Feeding at Discharge’	
	n = 3610		n = 3569		n = 3569	
Nagelkerke R-Square	0.32		0.33		0.327	
	OR (95% CI)	*p*	OR (95% CI)	*p*	OR (95% CI)	*p*
Birth weight (g)	0.999 (0.998–0.999)	<0.001	0.998 (0.998–0.999)	<0.001	0.998 (0.998–0.999)	<0.001
Male sex	1.355 (1.154–1.591)	<0.001	1.356 (1.153–1.594)	<0.001	1.355 (1.152–1.593)	<0.001
1 min Apgar score	1.033 (0.975–1.094)	0.275	1.031 (0.973–1.092)	0.301	1.033 (0.975–1.094)	0.275
5 min Apgar score	0.931 (0.864–1.002)	0.057	0.931 (0.865–1.003)	0.060	0.931 (0.864–1.002)	0.057
CRIB score	1.011 (0.976–1.048)	0.533	1.011 (0.976–1.048)	0.543	1.011 (0.976–1.048)	0.533
iMV	0.717 (0.539–0.954)	0.022	0.715 (0.537–0.950)	0.021	0.717 (0.539–0.954)	0.022
HFOV	2.103 (1.728–2.560)	<0.001	2.128 (1.746–2.594)	<0.001	2.124 (1.743–2.589)	<0.001
Duration of mechanical ventilation (hours)	1.002 (1.001–1.002)	<0.001	1.002 (1.001–1.002)	<0.001	1.002 (1.001–1.002)	<0.001
Surfactant therapy	1.021 (0.808–1.290)	0.861	0.999 (0.789–1.264)	0.992	1.009 (0.797–1.277)	0.94
iNO	1.655 (1.164–2.352)	0.005	1.717 (1.205–2.447)	0.003	1.741 (1.222–2.482)	0.002
PDA surgical/medical treatment	1.256 (1.064–1.483)	0.007	1.249 (1.056–1.447)	0.009	1.261 (1.067–1.491)	0.007
Late onset sepsis	1.646 (1.394–1.945)	<0.001	1.636 (1.383–1.936)	<0.001	1.634 (1.381–1.933)	<0.001
NEC	0.844 (0.658–1.084)	0.184	0.835 (0.649–1.074)	0.160	0.826 (0.642–1.063)	0.137
Exclusive HM at discharge	NA		0.758 (0.641–0.896)	0.001	NA	
Any amount of HM at discharge	NA		NA		0.770 (0.656–0.904)	0.001

Abbreviations. iMV: invasive mechanical ventilation, HFOV: high-frequency oscillatory ventilation, iNO: inhaled nitric oxide, PDA: patent ductus arteriosus. NEC: necrotizing enterocolitis; HM: human milk feeding.

## Data Availability

The data that support the findings of this study are available from the corresponding author, [R.P.], upon reasonable request.

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
