# Peer review of "Human Milk Feeding Is Associated with Decreased Incidence of Moderate-Severe Bronchopulmonary Dysplasia in Extremely Preterm Infants"

_children, 2023, doi:10.3390/children10071267_

Round 1

Reviewer 1 Report

The focus is interesting and having drawn from a national register allows you to have interesting numbers, but it is described in a confusing way and the result does not seem corroborated by specific data and there are contradictions.

The goal is poorly explained.

The study of associations is also unclear

The materials and methods do not define the sample recruited and the analyzes carried out. There is an endless list of of supposed independent variables and it is not known which they were related and how. Some reappear later, others do not. The structure of the study escapes.

The whole description appears approximate.

The results are not understood and should be rewritten schematizing them and starting from the objectives that are not clear.

There is the fundamental bias of the change of definition during the chosen period.

The theme concerning paracetamol has nothing to do with the context.

In discussion, the analysis of the literature appears poorly organized and not very effective.

The role and extent of the use of donated milk is not understood.

In the discussion they talk about studies without the bibliographic reference and without a precise order. The outline of the study escapes and how it came to the final sentence, expressed with a lot of emphasis.

The results do not seem to be supported.

Reviewer 2 Report

This is a database analysis. The main conclusion is breastfeeding at discharge decrease the incidence of moderate to severe BPD in extremely premature infants. I have several comments here:

1.     In figure 1, the BPD rates increase in recent years. The definition of BPD seems changed during the study period. If it is true, it is a major fallacy in this study because outcome definition should be consistent.

2.     The causal relationship of breastfeeding and BPD is possibly reversed. Infants with no or mild BPD could be discharged earlier, it is the reason why they have higher breastfeeding rate. Breastfeeding may be a surrogate marker for length of stay.

3.     Moreover, from the time sequence, breastfeeding at discharge happened after BPD formation. It could not be the cause of BPD.

4.     This article is well written. The case numbers are large. However, unclear outcome definition and reverse causal relationship rise a serious concern about the conclusion of this article.

Reviewer 3 Report

Human milk feeding could protect extremely preterm infants from bronchopulmonary dysplasia.
By Sergio Verd, et al.
The Authors described the positive effects of human milk in a retrospective study of extremely preterm newborns. The development of BPD is a complicate matter and a single therapeutic factor, i.e., human milk for audience, is not conceivable. The Authors suggest the administration and the positive effects of the untreated milk. This latter opens the chance of the gut pathogens’ microbiota colonization.
This paper sounds rightly done and it is of interest for audience, it is suitable for publication.

Reviewer 4 Report

In the paper entitled "Human milk feeding could protect extremely preterm infants from bronchopulmonary dysplasia", the authors concluded that human milk could prevent bronchopulmonary dysplasia in extremely preterm infants. This paper has too many limitations. The methods and discussion sections are poorly written. Authors need to describe the nutrition profile of the formula used for controls in detail. In the discussion section, the authors must focus on and discuss possible mechanisms by which Human milk feeding protects extremely preterm infants from bronchopulmonary dysplasia.

Minor: Better figure legends are needed (fig.1-3).

Round 2

Reviewer 2 Report

The authors have tried to answer my previous queries. However, the answers cannot satisfy me for accepting this manuscript. The reasons are:

1.     Although the authors answered my queries point by point, they haven’t told me where they changed their manuscript. The authors have to point out where they have modified their manuscript. There is an example at the submission page. 

2.     I haven’t seen length of stay in the regression model. How can the authors sate that they have controlled it?

3.     The title is “Human milk feeding could protect extremely preterm infants from bronchopulmonary dysplasia.” The authors also admitted that the causal relationship may be reverse. Breast feeding may be just associated with BPD. I don’t think the title is appropriate.

4.     Moreover, the conclusion “The results of the present study strongly support the role of any amount of human milk in preventing BDP in extremely preterm infants.” I cannot agree this conclusion based on the evidence they provided.

Reviewer 4 Report

The editor may accept  this paper in its present form

Author Response

Thanks for your review and your valuable decission.

Round 3

Reviewer 1 Report

focus on the triangle formed by: it doesn't make much sense, I would replace the word triangle

The classical criteria for BPD consider it as moderate if the infant is under supplementary oxygen with a FiO2 less than 0.3 and severe if the infant is under a FiO2 more than 0.3 or any positive pressure at 36 weeks of PMA.

If the classical criteria are named, the other criteria are also specified.

Male sex, HFOV, duration of iMV, iNO, treated PDA, and LOS were the other variables associated with moderate-severe BPD. It has been written before. It's a repetition

Finally, the temporal trends in human milk feeding were studied. It is a repetition. I would remove it.

Consequently, the rates of moderate-severe BPD in this group of preterm infants rise. This is a possible explanation for our sample's relatively low rate (43.7%) of moderate-severe BPD. You don't understand the meaning

Reviewer 2 Report

The authors made an attempt to address my inquiries, but I still have a few additional comments regarding their work:

1.     Regarding the association between breastfeeding and length of stay, it appears that breastfeeding may be acting as a surrogate for the length of stay in the study. To ensure that the results are not confounded, I suggest that the authors consider controlling for the length of stay in their statistical model. By accounting for the potential influence of length of stay, they can better isolate the effect of breastfeeding on the outcome of interest, reducing the possibility of any spurious associations.

2.     The authors mentioned that breastfeeding has the potential to protect against bronchopulmonary dysplasia (BPD). However, I find it puzzling that the incidence of BPD increases as the breastfeeding rate increases during the study period. This observation seems contradictory to the expected protective effect of breastfeeding. It would be helpful if the authors could provide a plausible explanation or discuss any potential confounding factors that might be influencing this finding. By doing so, they can help readers better understand the complex relationship between breastfeeding and BPD in the context of their study.

Overall, addressing these concerns will strengthen the validity and interpretation of the study findings.
